# Predictors of Treatment Response in Patients with Treatment-Resistant Depression: Outcomes of a Randomized Trial

**Karniza Khalid** [1,2,*], **Wei Chern Ang** [2], **Aimi Izwani Mohd. Nazli** [3], **Ruzita Jamaluddin** [2,3] **and Syed A. A. Rizvi** [4]

1   Special Protein Unit, Specialized Diagnostic Centre, Institute for Medical Research, National Institutes of Health, Ministry of Health Malaysia, Kuala Lumpur 50588, Malaysia
2   Clinical Research Centre, Hospital Tuanku Fauziah, Ministry of Health Malaysia, Kangar 01000, Malaysia; angweichern@moh.gov.my (W.C.A.)
3   Department of Psychiatry and Mental Health, Hospital Tuanku Fauziah, Ministry of Health Malaysia, Kangar 01000, Malaysia
4   College of Biomedical Sciences, Larkin University, Miami, FL 33169, USA
*   Correspondence: karniza.khalid@moh.gov.my; Tel.: +60-3-2616-2666 (ext. 2784)

**Abstract:** Objective: This report aims to compare the effectiveness between electroconvulsive therapy (ECT) and transcranial direct stimulation (tDCS) among patients with treatment-resistant depression (TRD) and the associated factors. Methods: This was a secondary data analysis of a randomized, controlled, open-label trial conducted from 1 July 2018, to 31 December 2018. The dataset of 90 patients included in the study was retrieved from Mendeley Data. Patients with TRD were randomized 1:1 into either receiving ECT or tDCS. A good treatment response was determined as an improvement from the Hamilton Depression (HAM-D) baseline score at a rate of more than or equal to 50% at the end of a 2-week intervention. A mixed-effect logistic regression was performed to explain the hierarchical data structure of predictors to successful treatment outcome. Results: The largest magnitude of change was consistently observed in the ECT treatment arm across both HAM-D score and the clinical global impression severity scale (CGI-S) scale ($p < 0.001$). Lower baseline HAM-D scores (OR: 0.72, 95% CI: 0.58, 0.92), lower baseline CGI-S scale (OR: 0.30, 95% CI: 0.17, 0.55), and ECT as the choice of treatment modality (OR: 14.0, 95% CI: 5.08, 38.58) independently predicted successful therapy among TRD patients, while modelling with multiple logistic regression determined that low socio-economic status (aOR: 20.01, 95% CI: 1.89, 211.47), ECT (aOR: 31.7, 95% CI: 6.32, 159.0) and a lower baseline CGI-S scale (aOR: 0.18, 95% CI: 0.06, 0.57) were significantly predictive of a positive treatment outcome among patients with TRD. Conclusions: ECT was more effective in alleviating depressive symptoms in TRD as compared to tDCS.

**Keywords:** depression; electroconvulsive therapy; transcranial direct current stimulation; treatment outcome; logistic models





## 1. Introduction

Depression is one of the most prevalent mental illnesses in the world, affecting an estimated 3.8% of the population, including 5% of adults and 5.7% of elderly over 60 years old [1]. The lifespan incidence rate of depression is noted to be higher among women than men, 20–25% and 7–12%, respectively [1]. In Malaysia, the national prevalence of depression is 2.3%, affecting approximately half a million of its population [2]. Major depression along with suicide risk and treatment-resistant depression (TRD) have a significant disease burden and incur substantial productivity costs to the nation's economy and society [3]; hence, they should be made into priority areas for community health benefits.

TRD is a subset of major depressive disorder and has been defined as the failure to respond to two or more antidepressants at a maximal dose (of at least 150 mg/day of

imipramine equivalent) for an adequate duration (at least 4 weeks) [4]. The classification of treatment-resistant depression is further complicated by the fact that different study settings have varying definitions of treatment response and success [5]. The Hamilton Depression Rating Scale (HAM-D) is the gold standard and a thoroughly researched method for assessing depression symptoms in clinical practice and clinical trials [6]. Clinical Global Impression (CGI) [7] is another most frequently used evaluation tool in psychiatry, and it is a brief clinical-rated measure of illness severity [8].

In one study, despite various classes of antidepressants, 20–30% of MDD patients still do not respond satisfactorily to the usual recommended dose of antidepressants and 15% develop chronic depression [4]. Of note, very recently, the working theory that brain chemical (serotonin being the major one) imbalance leads to depression has been questioned [9]. This, coupled with the fact that the use of antidepressants is associated with many serious side effects [10], warrants the pursuit of other treatment modalities.

The strategies that can be used in the treatment of MDD patients with partial or poor response to antidepressants include the optimization of the antidepressant, switching to a different class of antidepressant, using a combination of antidepressants, augmentation with atypical antipsychotics, lithium, antiepileptics or intranasal esketamine, and physical treatment such as electroconvulsive therapy (ECT), repetitive transcranial magnetic stimulation (rTMS), and transcranial direct stimulation (tDCS) [5,11,12].

Electroconvulsive therapy (ECT) is indicated for treating severe depression with poor response to antidepressants, particularly in patients with TRD, or patients with life-threatening conditions such as those with a refusal to eat and those who are highly suicidal [4,13]. The ECT procedure involves the delivery of small electrical currents through the brain, intentionally triggering a brief seizure that results in changes in the brain chemistry (particularly in the voxel-level whole-brain functional connectivity homogeneity (FcHo) in the right dorsomedial prefrontal cortex (dmPFC)) that may quickly reverse the symptoms of certain mental health conditions [14,15]. Unfortunately, many still refuse ECT owing to the stigma associated. Apart from that, ECT is also associated with an increased risk of cognitive impairment as compared to other physical treatments, mainly short-term retrograde amnesia, anterograde amnesia, and transient delirium [4,16]. Consequently, the search for alternative therapeutic options is crucial. tDCS is among the alternative therapy methods for which the efficacy in treating TRD is still being investigated.

tDCS is a non-invasive brain stimulation technique used to treat several neurological and psychiatric disorders. tDCS is based on the premise that individuals with major depressive disorder have hypoactivity in the left dorsolateral prefrontal cortex (DLPFC) and hyperactivity in the right DLPFC. It is a neuromodulator approach involving the direct application of a low-amplitude electrical current to a specified cortical region of the brain via electrodes attached to the scalp, thereby influencing neuronal networks [17,18]. Considering the numerous benefits of tDCS, such as its simple administration, mobility, low cost, and few side effects, it can be regarded as a potential alternative to ECT.

Hence, we aimed to provide a more extensive analysis from a publicly accessible dataset comparing the effectiveness between ECT and tDCS, based on HAM-D and CGI scores, and to determine the predictive factors to treatment response among patients with TRD.

## 2. Material and Methods

### 2.1. Overview of Study Design

We performed a secondary data analysis of a randomized, controlled, open-label interventional trial to study the effectiveness of electroconvulsive therapy (ECT) and transcranial direct stimulation (tDCS) among patients with treatment-resistant depression (TRD). The original study was conducted from 1 July 2018 to 31 December 2018 in an outpatient setting.

The dataset [19] used for analysis in this study was retrieved from Mendeley Data, a freely accessible, secure, cloud-based repository. The use of the dataset is in compliance

with Creative Commons Attribution 4.0 International Public Licensing and the terms of use. The retrieved dataset was adequately anonymized. Data quality and integrity were assessed through communication with the original authors whenever necessary. Data were collected and shared ethically. The original publication associated with the dataset was published in 2022 [9]. Ethical review has been exempted by the Medical Research and Ethics Committee (MREC) of the Ministry of Health (MOH) Malaysia as per the NIH Guidelines for Conducting Research in MOH Institutions and Facilities for the use of publicly available data. The study has been registered with the National Medical Research Register (NMRR) of the Ministry of Health Malaysia (NMRR ID-22-01642-U1A).

The trial had been originally approved by the internal ethics committee, Radianz Health Care (Ref.: 4/2018). There was no information on trial registration provided by the original authors.

### 2.2. Sample and Randomization

The dataset included a total of 90 patients identified to have TRD. The subjects included adults 18–60 years old, who met the criteria for MDD according to DSM 5 and TRD (defined as refractory response to at least two different pharmacological classes of antidepressants with maximal dosage, duration, and compliance for at least 6 weeks). Patients with bipolar affective disorder, schizophrenia or neurodevelopmental disorders (such as intellectual disabilities and other pervasive developmental disorders like autism spectrum disorder, attention deficit hyperactive disorder (ADHD), and communication disorder), patients with alcohol or psychoactive substance abuse, and pregnant or lactating women were excluded.

The study subjects were selected through stratified random sampling, while the treatment arm was determined using a block randomization technique and using a software-generated sequence. Informed consent was obtained prior to study enrolment.

The intervention (of either ECT or tDCS) was performed over the duration of 2 weeks. During the intervention, participants received a steady dose of antidepressants for at least four weeks prior to study enrolment and throughout the study period. Modified ECT was given thrice weekly (every Monday, Wednesday, and Friday). Prior to the procedure, patients were fasted overnight. During the procedure, a muscle relaxant and anaesthesia were given, with a stimulus dose titration performed at the first encounter to determine the seizure threshold. Standard brief-pulse ECT equipment and a bitemporal placement of the electrodes were used.

On the other hand, tDCS was administered on a daily basis (except on weekends). Patients were awake during the procedure, seated on a chair with an anodal electrode placed in the left DLPFC, and a cathodal electrode over the supraorbital region. Stimulation was given at 1–2 mA over 20 min. Six ECTs and 10 tDCS were administered to the participants during the stipulated time.

### 2.3. Data Collection

The data collated include the baseline socio-demographic details (i.e., age, gender, education level, employment and socioeconomic status, religion, and marital status), baseline and post-treatment depression scores using the Hamilton Depression Rating Scale (HAM-D) and clinical global impression severity (CGI-S) scale. Post-treatment HAM-D and CGI-S were taken at the end of the sixth ECT, while post-assessment for the tDCS arm was taken after the end of the tenth session. Routine follow-up was continued for these patients at the end of the study visit. Good treatment response was determined as an improvement from the HAM-D baseline score at a rate of more than, or equal to, 50% at the end of the 2-week intervention.

### 2.4. Statistical Analysis

Statistical analyses were performed using IBM SPSS Statistics (Version 26.0). Descriptive statistics were used to illustrate the baseline demographic and clinical characteristics of the study subjects.

A mixed-effect logistic regression was performed to explain the hierarchical data structure of predictors of a successful treatment outcome. Variables with $p < 0.25$ from simple logistic regression were selected for multiple logistic regression (MLR). Basic assumptions that were assessed included the independence of errors, linearity for continuous variables, the absence of multicollinearity, and the lack of strongly influential outliers. Omnibus tests of the model coefficients ($p < 0.001$) in MLR determined that the new model explains more of the variance in outcome; hence, it is significantly better than the baseline model. Nagelkerke's $R^2$ explained the percentage of variations in the outcome, while the Hosmer and Lemeshow test for goodness-of-fit suggested a fit model if $p > 0.05$.

## 3. Results

A total of 90 patients with TRD were recruited and randomized into either the ECT ($n = 46$, 51.1%) or tDCS treatment arm ($n = 44$, 48.9%). The mean age of the study participants was $40.8 \pm 11.75$ years, with a baseline HAM-D score of $19.1 \pm 2.04$ and CGI-S scale of $5.3 \pm 0.85$. Detailed sociodemographic and baseline clinical characteristics are shown in Table 1.

**Table 1.** Baseline sociodemographic and clinical characteristics of study participants (N = 90).

| Variable(s) | Overall n (%) | ECT ($n = 46$) n (%) | tDCS ($n = 44$) n (%) | *p*-Value [a] |
|---|---|---|---|---|
| Age (in years old) | $40.8 \pm 11.75$ | $38.2 \pm 11.25$ | $43.6 \pm 11.73$ | 0.026 *[b] |
| Gender | | | | 0.127 |
| Male | 38 (42.2) | 23 (60.5) | 15 (39.5) | |
| Female | 52 (57.8) | 23 (44.2) | 29 (55.8) | |
| Education level | | | | 0.745 |
| Basic | 28 (31.1) | 14 (50.0) | 14 (50.0) | |
| Intermediate | 32 (35.6) | 18 (56.3) | 14 (43.8) | |
| Advanced | 30 (33.3) | 14 (46.7) | 16 (53.3) | |
| Employment | | | | 0.020 * |
| Unemployed | 10 (11.1) | 5 (50.0) | 5 (50.0) | |
| Employed | 49 (54.5) | 31 (63.3) | 18 (36.7) | |
| Self-employed | 13 (14.4) | 2 (15.4) | 11 (84.6) | |
| Housewife | 18 (20.0) | 8 (44.4) | 10 (55.6) | |
| Socioeconomic status | | | | 0.924 |
| Low | 32 (35.6) | 17 (53.1) | 15 (46.9) | |
| Middle | 39 (43.3) | 20 (51.3) | 19 (48.7) | |
| High | 19 (21.1) | 9 (47.4) | 10 (52.6) | |
| Religion | | | | 0.929 |
| Hindu | 56 (62.2) | 29 (51.8) | 27 (48.2) | |
| Muslim | 15 (16.7) | 7 (46.7) | 8 (53.3) | |
| Christian | 19 (21.1) | 10 (52.6) | 9 (47.4) | |
| Marital status | | | | 0.920 [c] |
| Single | 8 (8.9) | 4 (50.0) | 4 (50.0) | |
| Married | 71 (78.9) | 35 (49.3) | 36 (50.7) | |
| Divorced/Widowed | 5 (5.5) | 2 (40.0) | 3 (60.0) | |
| Baseline HAM-D | $19.1 \pm 2.04$ | $19.0 \pm 1.95$ | $19.2 \pm 2.14$ | 0.565 [b] |
| Baseline CGI-S | $5.3 \pm 0.85$ | $5.1 \pm 0.89$ | $5.5 \pm 0.79$ | 0.071 [b] |

Note: [a] Pearson chi-square test of independence, [b] Independent *t*-test (described in mean $\pm$ SD), [c] Fisher's exact test. Abbreviation: ECT = Electroconvulsive therapy, tDCS = Transcranial direct current stimulation. * Statistically significant.

There were no significant differences with regard to baseline HAM-D ($18.98 \pm 1.95$ vs. $19.23 \pm 2.14$) and CGI-S scales ($5.13 \pm 0.89$ vs. $5.45 \pm 0.79$) between the ECT and tDCS treatment arms ($p > 0.05$). With regard to treatment response, the largest magnitude of change was consistently observed in the ECT treatment arm across both HAM-D score and the CGI-S scale following treatment (lower scores on both scales in the ECT treatment arm) (Figure 1).

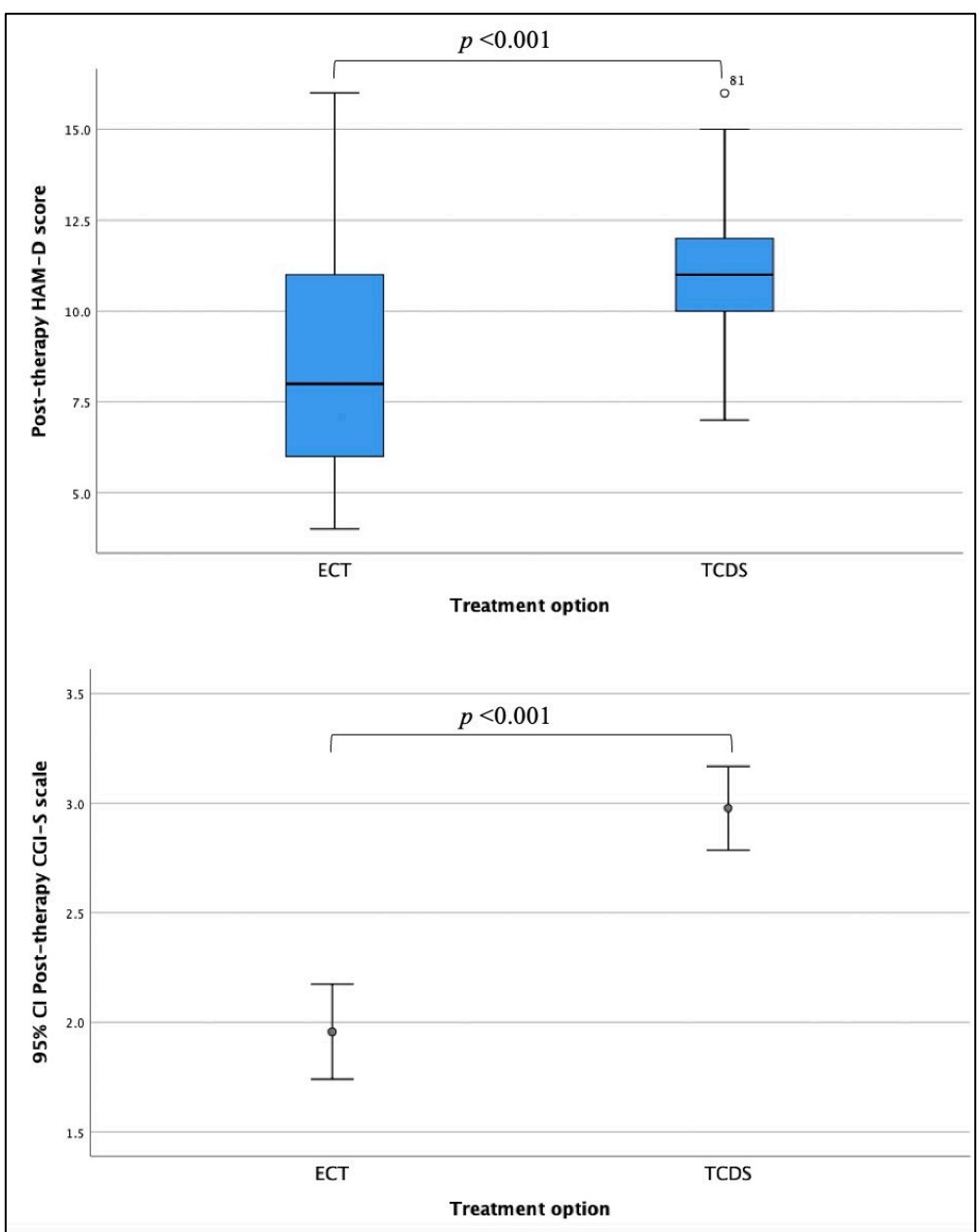

**Figure 1.** Comparison between ECT and tDCS treatment effects.

The baseline HAM-D scores, baseline CGI-S scale, and choice of treatment modality independently predicted successful therapy among TRD patients, while modelling with multiple logistic regression (MLR) determined that socio-economic status, treatment option, and baseline CGI-S scale were significantly predictive of a positive treatment outcome among patients with TRD (Table 2).

Our Omnibus tests of model coefficients ($p < 0.001$) in MLR determined that the new model explains more of the variance in the outcomes, and hence is significantly better than the baseline model. Nagelkerke's R2 explained roughly 64.5% of the variations in the outcome, while the Hosmer–Lemeshow test (HL test) of the goodness-of-fit suggested a fit model, $p = 0.518$ (>0.05). We found that the low- and middle-income groups had 20 and 3 times increased odds, respectively, of achieving a positive treatment outcome as compared to those with a high socioeconomic status. Similarly, patients subjected to ECT had almost 32 times increased odds of a successful treatment outcome as compared to those

with tDCS. We also found that an increase of 1 unit of the CGI-S scale would reduce the odds of a successful treatment outcome by 82%.

**Table 2.** Predictors of successful therapy among TRD patients.

| Variable(s) | Good Treatment Response *n* (%) | Poor Treatment Response *n* (%) | OR (95% CI) | *p*-Value [a] | Adj. OR (95% CI) | *p*-Value [b] |
|---|---|---|---|---|---|---|
| Age [†] | 39.2 ± 11.53 | 42.5 ± 11.85 | 0.98 (0.94, 1.01) | 0.176 | - | - |
| Gender | | | | 0.202 | | - |
| Male | 22 (57.9) | 16 (42.1) | 1.73 (0.75, 4.04) | | - | |
| Female | 23 (44.2) | 29 (55.8) | 1.00 (Ref.) | | | |
| Socioeconomic status | | | | 0.176 | | 0.025 * |
| Low | 20 (62.5) | 12 (37.5) | 2.86 (0.88, 9.25) | | 20.01 (1.89, 211.47) | |
| Middle | 18 (46.2) | 21 (53.8) | 1.47 (0.48, 4.53) | | 2.79 (0.43, 18.13) | |
| High | 7 (36.8) | 12 (63.2) | 1.00 (Ref.) | | 1.00 (Ref.) | |
| Education level | | | | 0.679 | | - |
| Basic | 15 (53.6) | 13 (46.4) | 1.01 (0.36, 2.84) | | | |
| Intermediate | 14 (43.8) | 18 (56.3) | 0.68 (0.25, 1.85) | | - | |
| Advanced | 16 (53.3) | 14 (46.7) | 1.00 (Ref.) | | | |
| Marital status | | | | 0.738 | | - |
| Single | 8 (57.1) | 6 (42.9) | 0.60 (0.10, 3.89) | | | |
| Married | 34 (47.9) | 37 (52.1) | 0.89 (0.11, 7.11) | | - | |
| Divorced/Widowed | 3 (60.0) | 2 (40.0) | 1.00 (Ref.) | | | |
| Employment | | | | 0.834 | | - |
| Unemployed | 5 (50.0) | 5 (50.0) | 1.00 (0.21, 4.69) | | | |
| Employed | 26 (53.1) | 23 (46.9) | 1.13 (0.38, 3.33) | | - | |
| Self-employed | 5 (38.5) | 8 (61.5) | 0.63 (0.15, 2.66) | | | |
| Housewife | 9 (50.0) | 9 (50.0) | 1.00 (Ref.) | | | |
| Religion | | | | 0.909 | | - |
| Hindu | 27 (48.2) | 29 (51.8) | 0.84 (0.30, 2.38) | | | |
| Muslim | 8 (53.3) | 7 (46.7) | 1.03 (0.27, 3.99) | | - | |
| Christian | 10 (52.6) | 9 (47.4) | 1.00 (Ref.) | | | |
| Treatment option | | | | <0.001 * | | <0.001 * |
| ECT | 36 (78.3) | 10 (21.7) | 14.00 (5.08, 38.58) | | 31.69 (6.32, 159.01) | |
| tDCS | 9 (20.5) | 35 (79.5) | 1.00 (Ref.) | | 1.00 (Ref.) | |
| Baseline HAM-D [‡] | 18.0 (2.00) | 20.0 (3.00) | 0.72 (0.58, 0.92) | 0.009 * | 0.93 (0.60, 1.44) | 0.734 |
| Baseline CGI-S [‡] | 5.0 (2.00) | 6.0 (0.00) | 0.30 (0.17, 0.55) | <0.001 * | 0.18 (0.06, 0.57) | 0.004 * |

Note: Abbreviation: CGI-S = Clinical Global Improvement Severity Scale, HAM-D = Hamilton Depression score. [†] Presented in mean ± standard deviation, [‡] Presented in median (IQR); [a] Simple logistic regression (presented in odds ratio and 95% CI), [b] Multiple logistic regression (presented in adjusted odds ratio and 95% CI) using Enter method, adjusted for age and gender. * Statistically significant.

## 4. Discussion

Our study aimed to assess the efficacy of ECT vs. tDCS in the treatment of TRD. Our study conclusively determined that treatment with ECT was associated with a greater treatment response, as observed with a significantly greater difference (reduction) in both HAM-D and CGI-S score as compared to tDCS. Despite the known effectiveness of ECT in various neuropsychiatric disorders, such as for manic episodes and mixed episodes in bipolar affective disorder, the specific mechanisms of how the seizure activity that is triggered during the process results in the amelioration of psychiatric symptoms is still not completely understood. Various biological theories have been suggested previously, including neurophysiological, neurobiochemical, and neuroplastic changes [17,20]. Unfortunately, the lack of research homogeneity renders inconsistent findings, and hence inconclusive inferences.

A recent paper has attempted to delineate different functional circuits that relate to effective antidepressant treatments together with the associated residual functional impairments [14]. The whole brain functional connectivity homogeneity (FcHo), located in the right dorsomedial prefrontal cortex (dmPFC), has been identified as the specific functional circuit supporting the neuroanatomical basis of the antidepressant effects of ECT [13]. Hence, future study is warranted to further validate the specificity of the improved functional circuit in the remission of depressive episodes to guide personalized medicine.

Interestingly, we also found that TRD patients from the lower socioeconomic status group had lower scores on the CGI-S scale, and, regarding treatment with ECT, they had better odds of a successful treatment outcome. This finding of lower socioeconomic groups having better chances of successful therapy contrasted with those of a recent

cross-sectional study among 44,805 patients seeking psychological aid for depression, which found that socioeconomically deprived patients required more treatment sessions to benefit from therapy [21]. In this instance, we suggest that cultural habit may play a role, such that people with scarcity mindset have a harder time taking in information and thus have less expectations [22]. This may contribute to greater subjective scoring when being assessed post-treatment in an attempt to please the clinician and maintain social acceptance [23]. Similarly, the CGI-S, which is a clinician-rated scale on the severity of mental illness [7], negatively predicts treatment outcome, whereby lower scores translate into a better treatment outcome. In other words, the lesser the severity of the depressive episode, the higher the chances of successful treatment [24].

The greatest magnitude of change in HAM-D and CGI-S scores was consistently observed among patients subjected to ECT as compared to tDCS. Patients subjected to ECT also had better chances at a successful treatment outcome. A meta-analysis conducted to determine the effectiveness of tDCS for MDD patients suggested the treatment effects may be observed beyond the intervention period [18,25], and hence they may not be immediately evident following treatment. Therefore, studies looking at the effect of these two treatment modalities may benefit from longer follow-up periods.

Despite the limitations, this pilot trial offers new insight with regard to treatment options in patients with TRD. In order to overcome the social stigma associated with ECT, its benefit should be communicated well to the patients and family members to ensure a proper informed decision is made in this coming era of personalized medicine.

### 4.1. Limitations of the Study

There are several limitations associated with this study, including, but not restricted to, the following: sample size; availability and concomitant use of various antidepressants; and hesitation and low compliance associated with treatment (logistic reasons, treatment time, and observation post-treatment).

### 4.2. Recommendations

We suggest for future studies to undertake longitudinal studies with a longer follow-up duration, and to be conducted within a larger sample size, in order to be able to conclude on the study hypothesis with confidence. In view of the evidence-based success of ECT in the management of TRD, we also suggest for future research to explore the potential of modified ECT delivery techniques to reduce the cognitive side effects that are commonly associated with ECT.

### 5. Conclusions

Our findings, after a secondary analysis of the available data, affirmed that ECT has a superior mood-stabilizing characteristic as compared to tDCS in the acute management of a depressive episode. Additionally, patients of a low and middle socioeconomic status, and with a lower CGI-S score, also had better chances of recovery.

**Author Contributions:** Conceptualization, K.K., W.C.A. and S.A.A.R.; methodology, K.K. and W.C.A.; software, K.K., A.I.M.N. and R.J.; data integrity, A.I.M.N. and R.J.; formal analysis, K.K. and S.A.A.R.; writing—original draft preparation, K.K., W.C.A. and R.J.; writing—review and editing, K.K., R.J. and S.A.A.R.; supervision, S.A.A.R. All authors have read and agreed to the published version of the manuscript.

**Funding:** This research received no external funding.

**Institutional Review Board Statement:** Not applicable.

**Informed Consent Statement:** Not applicable.

**Data Availability Statement:** Data are available in a publicly accessible repository.

**Acknowledgments:** The authors would like to thank the Director General of Health Malaysia for his permission to publish the paper.

**Conflicts of Interest:** The authors declare no conflict of interest.

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
