# Peer review of "Predictors of Treatment Response in Patients with Treatment-Resistant Depression: Outcomes of a Randomized Trial"

_2673-5318, doi:10.3390/psychiatryint4030025_

Round 1

Reviewer 1 Report

The article was reviewed and evaluated by me. I think it is necessary to focus on a few problems.

First of all, why have the results of this research, which was conducted in 2018, been delayed so long?

2. TRD identification is somewhat problematic. In particular, the duration and dose of antidepressants should be based on the last few international sources for TRD. For example, it would be more appropriate to use the term maximal dose rather than adequate dose in the definition.

3. Depression typing could have been included in the study title. (MDD instead of depression).

4. If the age range of the subjects were narrower, more accurate judgment could be made about response to treatment, resistance and 2 weeks of treatment.

In the 5th method section, the cathode placement could be detailed as right supra orbital or left supraorbital.

From these perspectives, I think that article writing should be reworked.

Sincerely yours

Author Response

As attached

Reviewer 2 Report

 But it is suggested to update the references and use more up-to-date documents in the discussion

Author Response

As attached.

Reviewer 3 Report

This paper aims to compare the effectiveness between electroconvulsive therapy  and transcranial direct stimulation among patients with treatment-resistant depression and associated factors. The topic is highly relevant in the treatment of Depression. The study is well executed, designed and reported. The research problem is properly formulated, and the aims are stated clearly. The research design is applicable, and the statistical methods are explained well. The conclusion is clear.

However there are some minor corrections to be done:

1) Six of the nineteen references are older than 5 years. Please update where possible.

2) Add a heading with the recommendations, which reflects the aims of the study.

Author Response

As attached.

Reviewer 4 Report

The content of the research should be improved:

1. The introduction fits the necessary information about the pathology but it is advisable to provide more information about the two treatment proposals. What is the basis of ECT?

2. The description of the methodology is timely, however the study design could be improved. The study variables and follow-up are limited.

3.  The discussion should be improved, it is possible to go deeper and provide more information related to the study. Could the proposed treatments be related to physical exercise? How?

Thank you

Author Response

As attached.

Reviewer 5 Report

This is a study done to compare the effect of two stimulation techniques as treatment for treatment-resistant depression (TRD). The techniques are tDCS and electroconvulsive therapy (ECT). The database was already available. The authors selected 90 patients from the database who underwent tDCS or ECT as treatment. Based on the results of analysis,ECT is more effective in alleviating depressive symptoms in TRD as compared to tDCS. I would suggest the following items to be included in the manuscript to improve its quality for being published:

 1-How were the MLR assumptions investigated?

2-  Repeated-measures ANOVA with stimulation (tDCS, ECT) * time (pre-stimulation, post-stimulation) for HAM-D and CGI-S, separately. Then ploting the pre/post values of measurments for the two groups in the same plot. 

3- Performing MLR model for ECT group and tDCS group separately, considering with baseline scores and  socio-economic status as predictors.

Author Response

As attached.

Round 2

Reviewer 4 Report

The authors' review is adequate

Author Response

Thank you for your feedback.

Reviewer 5 Report

Thanks for replying to some of my comments, however briefly.

Author Response

Thank you for your kind feedback.